# Sensory Stimulation in the NICU Environment: Devices, Systems, and Procedures to Protect and Stimulate Premature Babies

**DOI:** 10.3390/children8050334

**Published:** 2021-04-25

**Authors:** Francesco Massimo Vitale, Gaetano Chirico, Carmen Lentini

**Affiliations:** 1Psicologia Magistrale (LM-51), Clinical and Rehabilitation Psychology, Università Niccolò Cusano, 00166 Roma, Italy; 2Neonatology and Neonatal Intensive Care Unit, Children Hospital, ASST Spedali Civili, 25123 Brescia, Italy; 3Neonatal Pathology/NICU, Ospedale Civile Padova—AOPD, 35128 Padova, Italy; carmen.lentini@aopd.veneto.it

**Keywords:** soundscape, Vocalisation, NICU, music and music therapy, parent–infant interaction, MAMI VOiCE, Babybe, Babyleo, PAL^®^ (Pacifier Activated Lullaby System), Neoasis

## Abstract

Prematurity deprives infants of the prenatal sensory stimulation essential to their correct development; in addition, the stressful environment of the NICU impacts negatively on their growth. The purpose of this review was to investigate the effects of NICU noise pollution on preterm infants and parents. We focused on the systems and projects used to control and modulate sounds, as well as on those special devices and innovative systems used to deliver maternal sounds and vibrations to this population. The results showed beneficial effects on the preterm infants in different areas such as physiological, autonomic, and neurobehavioral development. Although most of these studies highlight positive reactions, there is also a general acknowledgement of the current limits: small and heterogeneous groups, lack of structured variable measurements, systematic control groups, longitudinal studies, and normative values. The mother’s presence is always preferred, but the use of music therapy and the devices analyzed, although not able to replace her presence, aim to soften her absence through familiar and protective stimuli, which is a very powerful aid during the COVID-19 pandemic.

## 1. Introduction

An estimated 15 million babies are born preterm every year, and this number is growing in the majority of countries. Globally, prematurity is the first cause of death among children under 5 years of age [1]. Many survivors face a lifetime of learning disabilities, such as visual and hearing problems and cerebral palsy, as well as feeding difficulties [2]. Prematurity also has as a negative impact on the family and health systems; parents have to cope with lifelong psychological and financial difficulties [3], and hospitals see their inpatient service costs rise significantly, both during the NICU stay and throughout their little patients’ infancy and early childhood [4]. Today, these problems have been exacerbated by COVID-19; indeed, many NICUs have limited parental access, with significant health and mental consequences for the neonate and their parents [5].

New technologies and noise reduction strategies (NAEP: noise awareness educational program) have recently been developed to rethink the NICU setting, providing an appropriate acoustic environment with sensors equipped with alarms, or special smartphone apps, and high-performance soundproofing materials have been implemented, as well as single rooms [6]. The sensory development of the preterm infants still advances even in the noisy environment of the NICU, thus the use of music therapy as a standard of care, combined with special devices, could help them to compensate for the abrupt separation from the mother womb [7].

The purpose of this review was to present and investigate the available technological devices, systems, and techniques, examined by appropriate scientific studies, used to promote premature babies’ optimal development and protect them from the excessive exposure to sounds and vibrations. The effect of such devices on parents and medical staff was also investigated, where possible. The study covered different areas of application of the vibration/sound treatments, many of them administered during music therapy sessions.

We investigated the new systems to control sound levels in the NICU and a new innovative incubator equipped with a stressor control. We also considered the effects of exposure to maternal sounds on the premature babies through different techniques and special devices, such as bone conduction, the Pacifier Activated Lullaby System^®^ (POWERS MEDICAL DEVICES, LLC, Boca Raton, FL, USA), the MAMI VOiCE System (© Associazione MAMI VOiCE, Brescia, Italy) and the Babybe^®^ System (© 2020 Natus Medical Incorporated, Pleasanton, CA, USA). Finally, we analyzed the effects of the Stochastic Resonance vibrating mattress and Neuromodulation on preterm infant’s apnea. The above devices and systems are all non-pharmacological and try to offer non-invasive, intuitive, and easy to use solutions that have the least interference with daily medical procedures and can safely and feasibly be used to soften the maternal absence. However, the live maternal voice as well as the physical and constant presence of the mother, in harmony with melody, modulation, and vibration is irreplaceable, because it is also thanks to the visual and vocal expressions of the caregiver that the baby learns [8,9,10].

Results also indicate that these procedures, despite some inconsistencies in their findings, show significant correlations with the fundamental role of the maternal voice, which represents a unique source of sensorial stimulus for preterm infants [11,12].

It is imperative to read this review knowing that the studies of the devices and techniques reported were applied to a heterogeneous population of patients with individual health conditions, in different environments, with different hospital guidelines, supervised by distinct nurses, doctors, and clinical operators at different times of the day; thus, it is neither possible to appropriately generalize these findings to a larger population, nor to realize a comparative review of the studies with the resulting analytical discussion.

## 2. Music Therapy

Since 2000, Music Therapy has been used in NICUs as a standard care to promote a healthier neurological, socio-affective development for the premature baby, and to support caregivers with a non-pharmacological treatment [13]. The soundscape of the NICU is full of potentially harmful stimuli, and in this setting, music therapy uses music (from live sessions or other technological devices) and specific recorded sounds, such as the mother’s voice [14] or her heartbeat [15], to mimic the ideal intrauterine environment, with empirically proven benefits on premature vital signs [16], cardiorespiratory stability, cortisol reduction [17,18], sleep-wake cycle, pain perception [19], and brain development (cognitive and sensory abilities) [20].

Thanks to the numerous studies in this relatively new field, music therapy beneficial effects have been validated by evidence from various disciplines such as neuroscience [21,22], physiology, psychology [23], and from studies of communicative musicality [24]. Music, as a sensorial and cognitive experience, suits perfectly both the fetus and the preterm infant’s predisposition to multisensorial experiences and, as a result, it stimulates and contributes to develop the baby’s sensory system [25]. The interventions [26] are carefully administered by specialized music therapists according to the infant age, state of health, hospitalization stage, and specific behavioral and physiological responses, which guide the music therapist’s choice of musical devices, type of sounds, and length of session. Music therapy interventions tend to be family-centered, helping parents to be more involved in the care of their baby [27,28,29], thus increasing their sense of empowerment through training musical sessions and counselling, even after the baby’s discharge [30]. Moreover, music therapy supports parents in rebuilding the relationship with their baby through intuitive vocal interactions, helping them to better cope with anxiety and stress [31,32,33].

Such science is also suitable to be part of an integrated approach when combined with other types of treatments, such as Kangaroo Mother Care, infant massage, and gentle handling [34]. Music therapy proven beneficial effects are evident in the short term, but further longitudinal studies are needed to confirm these findings, particularly for the infant’s neurodevelopment [35].

## 3. Devices Controlling Environmental Factors

### Noise Pollution

Ensuring control of risk factors in an NICU is of paramount importance. Brain development is affected by early auditory experiences; noise may be associated with accelerated heart rate or bradycardia, apnea, decreased oxygenation, increased muscle tension, blood pressure and intracranial pressure, sleep disturbance, agitation, and hearing and speech deficits. Studies show that a stay in the NICU for more than 4 days is a risk factor for hearing loss [36]. It exacerbates children’s energy expenditure, induces physiological instability, and may affect growth and development. In addition, recent studies reveal that the intensity of sound not only has a negative effect on the infant, but can also interfere with caregivers’ communication and job performance. The most currently used means of reducing such noise include single-family rooms and ear protection, which have had limited success. Importantly, basic re-education dedicated to the entire care team should target behavioral changes to decrease sound levels in the NICU [37], taking into consideration recommended developmental care models such as “NIDCAP” or “NAEP”. These programs include basic tips such as “speak softly and in low tones”, “wear soft shoes”, “do not use incubator tops as table surfaces”, “close porthole doors carefully”, “respond promptly to alarms”, and “limit use of personal radios”. In addition, it must be remembered that there is a strong relationship between the frequency of the auditory stimulus and the frequency of the tactile stimulus, which is the result generated by physical processes that generate stimuli. Such vibrotactile perception can be integrated with other senses (e.g., vision and hearing) to form multimodal events that are productive for the young patient [38]. Therefore, extreme care must be taken not to overwhelm the child’s immature neurological system [39].

To date, there are a number of tools useful for monitoring and/or modulating the sound level in NICUs. Some authors have investigated sound monitoring with a smartphone application, useful for identifying modifiable environmental factors in different forms of NICUs design. In this study, minimum, maximum, and peak decibel recordings were obtained using the Decibel X App, and the presence of noise sources was recorded in each space at three NICUs over a 6-month period (December 2017 to May 2018). Recorded sound levels exceeded recommended limits, and sound intensity varied by day/night shift, with higher daytime exposures in all units [40]. The app may be useful in quality improvement efforts to minimize environmental sound exposure. The design of easy-to-access and easy-to-use applets could facilitate and simplify the control of a difficult-to-contain factor such as sound and its vibrations.

Another study, conducted over a period from September 2002 to February 2003, examined the control of sound and sudden peak noise frequencies within incubators and infrared heated beds, both when combined with ventilation, monitored continuously for a period of 7 days before and one month after the introduction of a light system. The alarm uses microphone and amplifier circuitry to convert ambient noise (the sum of the acoustic energy of a human voice, equipment noise, air conditioner, etc.) to electrical potential. This electric potential, when above the set point, is modulated by the alarm signal of the red-light emitting diode (LED lamp). In this system, the alarm is activated when the sound level reaches 65 Db [41]. The noise sensor light alarm could effectively reduce the sound level and peak episodes in neonatal intensive care units, relieving the noise stress for critically ill infants.

Recently, a non-contact active incubator noise control device Neoasis™ (©2018 Invictus Medical, Inc., San Antonio, TX, USA) (Figure 1) was designed and evaluated; experiments were conducted in the Neonatal Intensive Care Unit simulation lab at the Children’s Hospital of San Antonio, using bedside ICU equipment to determine if it could effectively reduce noise exposure to infants inside a thermal crib. In the simulation center, a series of clinically appropriate sound sequences (exactly 10) were generated with medical devices, monitors and fluid infusion devices, a female and male voice, hospital air handling systems, and mechanical device sounds, clinically realistic for testing; each sequence was repeated for five trials. A newborn mannequin equipped with a microphone was placed inside an incubator, and measurements were made with the Neoasis™ turned off and on. The system consists of a control unit and an external noise sensor, both located outside the incubator, and two speakers and a residual noise sensor located inside the incubator. Active noise control uses the phenomenon of incident wave summation; if one incident wave is out of phase with the other, the waves cancel each other. The device measures sound waves outside the incubator, models what the sound waves will be after passing through the incubator wall, and generates a sound wave that is out of phase with the modeled wave; finally, a residual noise sensor inside the incubator provides the data for the system to converge to an optimal solution. This study demonstrates that an active noise control device provides noise attenuation within an incubator in a real-world simulation environment. The active noise control device reduced sound pressure levels for some alarm sounds by up to 14.4 dBA (a 5.2-fold reduction in sound pressure) at the primary frequency of the alarm tone. Frequencies below the 2 kHz octave band were attenuated more effectively than frequencies at or above the 2 kHz octave band. Background noise levels below 40 dB were not substantially affected by the active noise control device [42]. The active noise control device further reduces noise within infant incubators; the safety of the device and potential health benefits of a quieter environment should be tested in a clinical setting.

## 4. Sounds and Vibration Devices

### 4.1. Babyleo^®^ TN500

The Babyleo^®^ TN500 (© Drägerwerk AG & Co. KGaA, 2021, Lübeck, Germany) (Figure 2) is an incubator with acoustic and luminous comfort made by Dräger and produced with the help of simple 2D or 3D models to counteract the impact of both the acoustic and luminous factors. Its creators call it the best alternative after the womb. The chosen device was designed with the child’s development in mind, has low operating noise levels, and features noise and lighting monitoring within the patient compartment. It is currently in use in several NICUs around the world; in Italy, for example, it is present in the Department of Neonatal Pathology in Padua. By its design, you are aware of the exposure to potentially harmful stimulation and can react to reduce these sources of stress. A built-in safe auditory stimulation feature allows you to use a recording of the mother’s heartbeat or voice to calm or stabilize the baby. The device may support mother-infant bonding through distal vocal contact [43]. A small speaker then silently plays the recording inside the device. It would seem that incubators should be designed to improve acoustic comfort.

### 4.2. Bone Conduction

Bone conduction [44,45] takes advantage of the body’s natural ability to conduct sound in the transmission of acoustic vibrations through the head bones; the vibration, emitted from various devices, such as headphones, earphones, glasses, and bracelets (headband) with Velcro, bypasses the eardrum and transmits sound directly to the inner ear, which transforms it into a message for the brain [46,47]. In particular, bone conduction implants can help adults and children with hearing loss, and there was an Italian study, conducted in 2014, on the transmission via bone conduction of maternal voice to premature infants in an unnatural environment such as the NICU [48]. The study starts from the way the fetus receives sounds in the maternal uterus; the maternal voice reaches the fetus directly from the inside, propagating through the organs, in particular through the skeletal system. From the larynx, it goes down along the spinal column and reaches the pelvis that acts as a resonance chamber; finally, the sound pressure in the amniotic fluid produces vibrations of the skull that are transmitted into the cranial cavity, and from there into the fluids of the cochlea. The longitudinal, exploratory, control study assume that exposure of preterm infants to maternal voice via bone conduction mimics the modus operandi of the fetal auditory system. The aim, therefore, was to investigate the effects of this procedure on the autonomic and neurobehavioral development of infants, and to test the hypothesis that preterm infants exposed to maternal voice through bone conduction show better autonomic parameters during the period of intervention, and better neurobehavioral performance during the first 6 months of life, than the control group; the 71 preterm infants compared to a control group, were subjected to a neurofunctional assessment at 3 and 6 months of corrected age. Recorded maternal voice, adhering to AAP-recommended safety levels, was delivered during three sessions per day for twenty-one days, each followed by neurobehavioral assessment. A bone conduction transducer that converts an electrical signal into magnetic vibration applied to the wrist of each child was used in the intervention group. The maternal voice was transmitted through the transducer while the mother was reading passages from The Little Prince, and the voice was filtered to make it as similar as possible to that heard in utero. The results were positive; the preterm infants in the intervention group showed, as compared to the control group, a decrease in heart rate, greater stability in skin color, better performance of visual attention, and quality of movements, obtaining higher scores in the neuro-functional assessment at 3 months, while there was no difference in the two groups at 6 months.

Exposure to the maternal voice through bone conduction may therefore support the autonomic and neurobehavioral development of premature infants, at least in the first three months. Because this is the only study, further research and evaluations on the use of bone conduction in the NICU are needed, especially because of the problems related to the use of headphones on patients who are neurologically immature or affected by pathologies.

### 4.3. Pacifier Activated Device

Due to the immaturity of the nervous system, sucking difficulties also accompany the premature infant for a long time. Often babies are uncoordinated; they have difficulty in alternating sucking, swallowing and breathing movements and take a long time to learn them. These oromotor difficulties prevent proper growth, lead to long hospital stays and re-hospitalizations, and increase the stress of both the newborn and its parents [49]. Numerous researches have been initiated with the aim of solving these problems, including special tools such as the Pacifier Activated Lullaby System (PAL^®^) (© Powers Medical Devices, LLC, 2019, Boca Raton, FL, USA) [50,51,52] (Figure 3).

The musical lullaby is a medical device, used during music therapy sessions in the NICU, that reproduces the mother’s voice; it was created in 2000 by Dr. Jayne Standley, professor of music therapy at the University of Florida, with the aim of teaching preterm infants to coordinate their oromotor skills by strengthening the muscles of the mouth and throat, to allow them to be able to adequately feed, both at the breast and at the bottle [53]. A prospective, randomized control study [54] aimed to verify whether the use of pacifier-activated music players, playing the maternal voice during non-nutritive sucking (NNS), could help to develop oromotor skills and have a positive impact on preterm feeding. A music therapist randomly administered 15-min PAL^®^ sessions per day, for 5 consecutive days, to preterm infants selected by gestational age, nutrition status, and maternal voice recording as well as non-voice PAL^®^. The ergonomic device, which is lightweight and equipped with a sensor, is connected by a cable to a speaker. The speaker has an LCD screen that allows staff and parents to monitor progress and select modes of delivery; it is then placed inside the thermal cradle above the head or beside the newborn. When the baby sucks correctly the sensor of the pacifier sends the signal to the speaker that rewards his effort by playing a lullaby previously registered by the mother. During the sessions, the values concerning timing, quality and quantity of sucking, salivary cortisol levels, and growth parameters were noted. The results were encouraging; the intervention group significantly improved and increased the baby’s sucking capacity, stress levels were reduced, as were intubation and hospitalization times and the time needed to switch to exclusive oral feeding. Compared to the control group, there was no significant increase in weight, but this can also be explained by the short period of administration. The musical pacifier with the mother’s voice not only improved the nutritional capacity of the premature baby, but also involved the parents, who strengthened the bond with their baby and were less anxious and stressed as they were able to interact with the infant by holding him or stroking him in the crib during the music therapy session.

However, questions remain about the treatment of the most at-risk late preterm infants analyzed, while the fundamental importance of the maternal voice in stimulating the development of preterm infants emerges strongly. A follow-up study [55], complementing the previous one, observed the first year of life of the same preterm infants with the aim of verifying that the optimal acquisition of oromotor skills before discharge from the NICU may represent a solid basis for the acquisition of further and more complex skills, as well as a prevention of the development of complications and possible readmission to hospital. The researchers used the medical records and nutritional history of the young patients and sought the cooperation of some of the parents (72 of 94) from the 2014 study. Parents were contacted several times a day by telephone to answer special questionnaires. The processing of the data thus confirmed the effectiveness of PAL^®^, that one year later did not preclude the achievement of more complex oromotor skills, nor increased the risk of nutrition-related diseases that were related to any rehospitalizations. In addition, subjects in the intervention group no longer needed the maternal vocal stimulus to feed. The use of PAL^®^ in the NICU is therefore confirmed to be an effective intervention to improve the nutritional capacity of premature infants in the first year of life, though further studies should be undertaken regarding its long-term effects.

### 4.4. Babybe System^®^ (“Be with My Baby”)

In 2013, the Chilean engineer Dr. C. Anabalon, together with some neonatologists, studied a safe solution that wanted to recover the fundamental bond with the mother by imitating kangaroo care [56], allowing the premature baby to feel safe as if it was leaning against the mother’s belly, comforted by familiar sounds. This is the system called Babybe^®^ (“Be with my baby”) (© Natus Medical Incorporated, 2020, Pleasanton, CA, USA) [57], which, through a gel mattress placed in the cradle, is able to send the newborn the maternal heartbeat and breathing (Figure 4).

The maternal voice, spoken, sung, and read, can be sent to the mattress using a smartphone or MP3 device connected to a control module outside the thermal cradle. In this way, preemies can listen to them without the maternal presence, which may provide a useful tool in inevitable contexts of separation from the parents [58]. The procedure is family-centered because it involves the parents, especially the mother, who regains her role by reconnecting with her baby [59]. The pilot study was conducted in Chile (Hospital SanBorja Arriaran in Santiago) in 2013 and lasted two years, although scientific investigations have yet to be fully developed, the apparatus has been reported to be associated with favorable outcomes for both parents and babies; today, the Babybe system is in use in several hospitals: in the Netherlands (Maxima MC Hospital Eindhoven), in Germany (Mannheim University Hospital), in Italy (Hospital S. Bortolo in Vicenza) [60], and in the USA (Brigham and Women’s Hospital Boston) [61]. The Babybe^®^ technology [62] is based on three components: a turtle-shaped pillow, a control module, and a mattress to be inserted in the thermal cradle. The turtle-shaped plastic and gel pillow is equipped with sensors; the mother places it on her belly in the same position as the kangaroo care. The turtle weighs approximately 1.5 kg, is easy to use, and informs the mother that it has been correctly positioned by lighting up blue. The turtle (“mode” stage) contains sensors that detect, record, and wirelessly send real-time breath movement and heartbeat to a control module. The control module has four modes: “the automatic” mode automatically sends both the breath and the maternal heartbeat, the “mother mode” transmits the maternal breath in real time, the “mother replay mode” plays for seven minutes the maternal breath and the heartbeat of the last session, and finally the “audio playback mode” that, through cell phone or player, transmits the voice and music tracks. The process control module (“process” phase) receives the signal from the turtle, stores it, and uses it to control the mattress to which it is connected through air tubes. Breath movement and heartbeat (more percussive) will be sent to the mattress via air, while music and mother’s voice is sent as sound. The bionic polyurethane gel mattress is non-toxic, biocompatible, soft and elastic to the touch, and has an irregular surface whose texture imitates human skin. Inside it has pneumatic pumps (two “lung bladders’’ and a “heart bladder”) that inflate and deflate according to the mother’s respiratory movement and the percussive rhythm of the heartbeat, and the infant can be placed in a prone or supine position [63]. Babybe^®^ has produced positive results, both for the young patients and their parents, and for the hospital; in premature infants, the restoration of the bond with the mother has led to an improvement in respiratory level, a reduction in apnea events, and a lowering of cortisol levels, combined with weight gain, that has reduced the time of hospitalization. On the hospital side, the use of Babybe^®^ has brought about a decrease in costs and the workload of healthcare staff and an increase in patient turnover. In November 2020, Babybe^®^ GmbH was incorporated by “Natus Medical Incorporated©”, which has committed to large-scale production of the device by 2022 [64].

### 4.5. MAMI VoICE System

Along the same line of thought, in order to comfort babies born prematurely and to reduce, as much as possible, the feeling of separation from their mother during their permanence in the incubator, the MAMI VOiCE system (© Associazione MAMI VOiCE, 2018, Brescia, Italy) [65] (Figure 5) was conceived in 2005, thanks to the Brescia architect Alfredo Bigogno. The system is built and assembled in Italy by “© Associazione MAMI VOiCE” and has a medical certification “class 1”.

This system is connected to the incubator without the need to introduce equipment, without interfering with the fundamental work of medical staff, and without additional electromagnetic fields disturbing the baby. The system reproduces, in the shape of sound and vibrations, the recording of the mother’s voice (or appropriate audio/musical tracks), thanks to the presence of a vibro-transducer that creates “sound vibrations”, which harmoniously propagate through the whole incubator and the premature baby, thus stimulating its sensory system. The vibro-transducer is connected to an amplifier that is powered by batteries (especially shielded); in this an USB drive is inserted (with the possibility to add more) containing the recordings, chosen by the parents in total autonomy. The amplifier is equipped with an adjustable volume, at preset dB (preset at max volume, not exceeding the limits recommended by the American Academy of Pediatrics guidelines), with four specific buttons: one to turn on/off the system, one to start the recording (choosing between infinite loop or playing with pauses), and two for adjusting the volume.

The system has been used, and is currently under experimentation, in several hospitals distributed throughout Italy, Palestine, and Israel. A prospective, controlled study, carried out at the ASST Spedali Civili Brescia, in the department of Neonatology and Neonatal Intensive Care, from December 2013 to December 2015, investigated specifically the ability of this device to modulate pain perception [19]. Forty preterm infants were enrolled and randomized to listen or not listen to a recording of the mother’s voice during a painful, routine heel lance for blood collection. Changes in the preterm infant’s pain profile, heart rate, oxygen saturation, and blood pressure during the procedure were compared by analysis of variance. Possible side effects of apnea, bradycardia, seizures, and vomiting were also recorded. Results from both groups showed marked increases in Premature Infant Pain Profile (PIPP) scores and decreases in oxygen saturation during the procedure, but infants in the treatment group had significantly lower PIPP scores (*p* = 0.00002) and smaller decreases in oxygen saturation (*p* = 0.0283), and no significant side effects were observed. Therefore, the use of recorded maternal voices to limit pain in premature infants undergoing heel lancing procedures appeared to be safe and effective.

Another filed application of MAMI VOiCE was conducted by one of the co-authors, over a 2-month period of observation in the NICU of Santa Chiara hospital in Pisa (Italy) [66] for the conclusion of a semi-experimental thesis. This was a monocentric comparative observational study that focused on two periods of observation of the newborn’s behavior and vital parameters. The premature infants were observed while listening to the melody of the maternal voice recorded and transmitted through the device MAMI VOiCE, in the moments when they were subjected or not to diagnostic and therapeutic procedures. This study was applied in order to evaluate if the melody produced differences primarily on the sensory saturation and neurobehavioral stimulation of premature infants. The system in this study was used as a real tool of sensory saturation an innovative method of non-pharmacological neonatal analgesia applicable during the many maneuvers to which they are subjected. The use of maternal voice, and the resulting sensorial stimulation, helped to limit pain in premature infants and stabilize vital parameters such as heart rate and oxygen saturation.

The scientific literature of the last decade has shown how both the fetus and the newborn are able to perceive, decode, and store painful stimuli by responding to them with a series of psycho-emotional and behavioral reactions. In this way, we try to respect the basic principle of SS; that is, the gate theory already developed by Melzack and Wall [67] in 1965, acting on the fact that the brain, activated, distracted, and stimulated, according to the gate theory, acts as a filter that does not let pass all the inputs (nociceptive and tactile) because they compete with each other. The purpose of sensorial saturation is basically to distract and comfort the newborn; that is, to consider a premature baby as if it was a “big” baby when it suffers a trauma, reassuring it first, distracting it during, and comforting it after the procedure. This system turned out to be compatible with all other techniques of sensorial saturation. It is also capable of attracting the infant’s attention with positive multisensory stimuli that avoid physical contact that is not strictly necessary, acting as a sort of filter capable of reducing or excluding the input of discomfort and pain.

For children hospitalized in intensive care units, challenged by the difficult task of early adaptation to life outside the uterus, hearing and perceiving through a device the vibrations of the mother’s voice when she is not present can become a supporting procedure when the actual maternal presence is not possible, once robust empirical evidence could support the initial findings.

### 4.6. Stochastic Resonance

According to the basic principle of stochastic resonance [68], the application of a modest amount of random vibration to a complex biological system, such as the human body, increases the sensitivity of that system. Stochastic resonance has a wide range of applications through different disciplines; for example, in the case of apnea episodes, physicians treat premature infants with caffeine or respiratory support systems; these are effective and widely used treatments, but they do not completely resolve the problem. Apnea can cause developmental delays, long-term damage, and even death; in this regard, soft vibrations could alleviate apnea in premature infants [69].

An example (Figure 6) of the use of stochastic resonance can be found thanks to collaboration between the Wyss Institute for Biologically Inspired Engineering at Harvard University, the Beth Israel Deaconess Medical Center (BIDMC), and the University of Massachusetts Medical School (UMMS). A clinical trial showed that stochastic resonance stimulation can be used to treat premature infants with apnea, bradycardia, and oxygen desaturation, with positive outcomes. It was also intended to confirm the validity of previous research [70] that had already demonstrated how stochastic vibrotactile stimulation, applied to the total surface of the mattress, could stabilize the respiratory parameters of preterm infants. A randomized crossover study [71] was conducted from April 2012 to July 2014; the 36 involved premature infants (gestational age less than 36 weeks) had had at least one documented event of apnea, bradycardia, and/or desaturation. Infants placed on the special mattress received 30-min interventions for a total of 3 to 4 h, alternating on-off intervals of stochastic resonance stimulation. The mattress provided them with stimulation based on a gentle vibration similar to a gentle massage; all the while monitoring heart rate, respiratory rate, and oxygen saturation levels for each preemie, acting as its own “control”; they then compared the periods of on and off stimulation.

The researchers employed technology created by the Wyss Institute, which uses stochastic resonance imaging to prevent apnea in premature infants. It is composed of three elements: “Bedside data acquisition software” (developed by ©MediCollector) [72], connected to the monitors of the patients, is able to record the complete flow of parameters such as respiratory ones; these data can be viewed in real time, then recorded and sent to other applications in order to help the analysis of critical episodes and facilitate the therapies. The second element is an algorithm [73] that analyzes the succession of signals to predict apnea episodes, and finally the third element consists of a special therapeutic mattress (“vibrating mattress”) that is set to vibrate and restore regular breathing as soon as an apnea episode is announced. Compared to the 2009 study, care was taken to prevent the vibrations in the area corresponding to the infant’s head in order to avoid harmful effects on the still-developing brain. The stochastic resonance stimulation on three events: apnea, bradycardia, and oxygen desaturation, showed promising results; in fact, apnea events were reduced by 50%, bradycardia events intensity by 20%, and oxygen desaturation events by 20% to 35%.

### 4.7. Neuromodulation

The principle [74] behind neuromodulation has been known for some time; it is based on the close correlation between limb movement and breathing patterns, and consequently the influence of movement on increased breathing.

Researchers from the University of California published, in 2016, a study on the use of “neuro-modulation applied to the proprioceptive afferents of the limbs” of the premature babies, with the purpose of finding a valid alternative to the current procedures and therapies, aimed at reducing the events of apnea, oxygen desaturation, and bradycardia [75] (Figure 7).

Because the mechanical movement of extension and flexion is impossible to obtain in a newborn, the proprioceptive fibers conducting kinesthetic signals from the limbs to the areas of the brain that govern the movement, and that reflexively couple with those responsible for breathing, have been stimulated and activated by vibration [76,77]. Therefore, limb movement has been simulated by a slight vibration of proprioceptors in the premature infant’s hand or foot. The vibration device consisted of two elements: a stimulation device, containing a low-voltage battery that powers a vibration motor similar to that of cell phones, and “pucks”; the pucks were placed on the proprioceptive fibers of the palm or wrist of one hand and one ankle, or the sole of the foot of the premature infant, and always on the same part of the body. Mild vibration (0.3/128 Hz) was administered for a total of 24 h in on-off and off-on sequences of 6 h, always under continuous monitoring and recording of cardiorespiratory parameters. Neuro-modulation significantly lowered the number and duration of apneas, episodes of intermittent hypoxia, bradycardia, and oxygen desaturation. The use of neuro-modulation, through mechanical vibration stimulation of proprioceptive afferent membranes, has proved to be an economical, simple to use, and non-invasive technique compared to traditional ventilation techniques, which can damage the infant’s lungs. It also did not affect the quality of sleep of the premature infant, with some probability that it may reduce sleep disturbances in the future. Early neuromodulation intervention and treatment of apnea, oxygen saturation, and bradycardia events has the potential to improve long-term neurodevelopmental and pulmonary outcomes, although positive findings need to be confirmed by further studies.

## 5. Discussion

The NICU is a complex, highly technological setting, intended to provide the premature infants with the best standards of care, as well as promoting family-centered treatment by several steps, as music therapy sessions. The findings of this review show the potential benefits of implementing technologies, devices, and systems that control noise and promote the use of maternal sounds, vibrations, and music in the premature baby’s development. The devices and systems analyzed have generally reported positive results, such as improved infants’ physiological, neurodevelopmental, and behavioral parameters in the short term and a more secure parental bond, thus decreasing the stress levels of the family system. As stated in the introduction, these findings are limited by the setting and population analyzed; there is also a lack of longitudinal studies and clinical trials that could further support the role of these devices and technologies. Although we have seen that there is a correlation between external stimuli, such as sounds and vibrations, and the development and wellbeing of premature babies, there is no scientific evidence on the existence of a causal link between a specific frequency and a behavioral/physiological response or change. It will be useful for future medical and psychological assessments to have a record of these interventions to better understand if there are long-term effects. In this regard, it could be helpful to create a baby surrogate, made of hydrogel [78] (or some other material that mimics the density of human tissues and organs) and harder materials for the skeletal system, to be placed in an incubator. Vibro acoustic sensors [79] could be placed inside the surrogate body in strategic positions to better emulate what the baby can feel. There could also be additional sensors inside and outside the incubator to detect all possible stimuli. Each NICU presents a different environment, so a surrogate baby could be useful to measure the differences in sounds and vibrations at different times of the day; the data obtained could then be shared to improve the construction materials of the surrogate or to have more realistic values that could help improving the safety of the NICU environment. An important limitation of these studies is represented by the absence of an in-depth analysis of the neural activity that might confirm a causality relation between the positive outcomes and the devices and technologies utilized.

## 6. Conclusions

The use of maternal sounds, vibrations, music, and sensory protection protocols for premature infants hospitalized in the NICU could contribute to their optimal development, although further studies with larger sample sizes and long-term assessments are needed to confirm the above findings. Future clinical trials could help compare short- and long-term health effects with different devices and systems in order to create an effective recommendation for implementation guidelines. The maternal multifactorial sensory stimulations and complex interactions (voice and singing, breast feeding, face sighting, touch, massaging, kangaroo mother care, smell, warmth), and their unparalleled effects on the infant normal development, are impossible to imitate or reproduce.

However, further research should explore if the above “devices” may maintain a continuous sensory stimulation during the maternal absence, and if such additional experience were confirmed to be overall beneficial for the infants’ development and their parents’ well-being. these devices may be explored as a way to augmenting, rather than replacing, the parents’ involvement in NICU. In conclusion, the reviewed studies show that the most fertile discussions and findings are the results of multidisciplinary efforts, which involve different disciplines such as medicine, music, architecture, physics and engineering, further stimulating more comprehensive and collaborative discoveries.

## Figures and Tables

**Figure 1 children-08-00334-f001:**
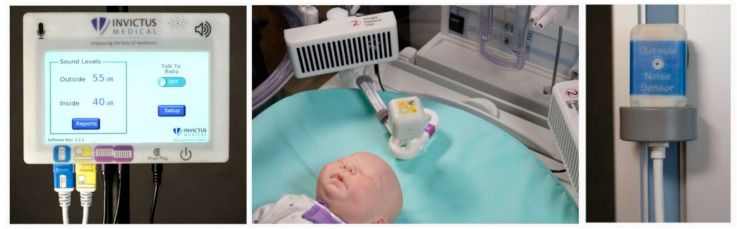
Neoasis™ system: on the left is the control unit with the external noise sensor, in the center are the speakers and the internal noise sensor, on the right is the additional “outside” noise sensor. http://www.invictusmed.com/neoasis/ (accessed on 6 February 2021).

**Figure 2 children-08-00334-f002:**
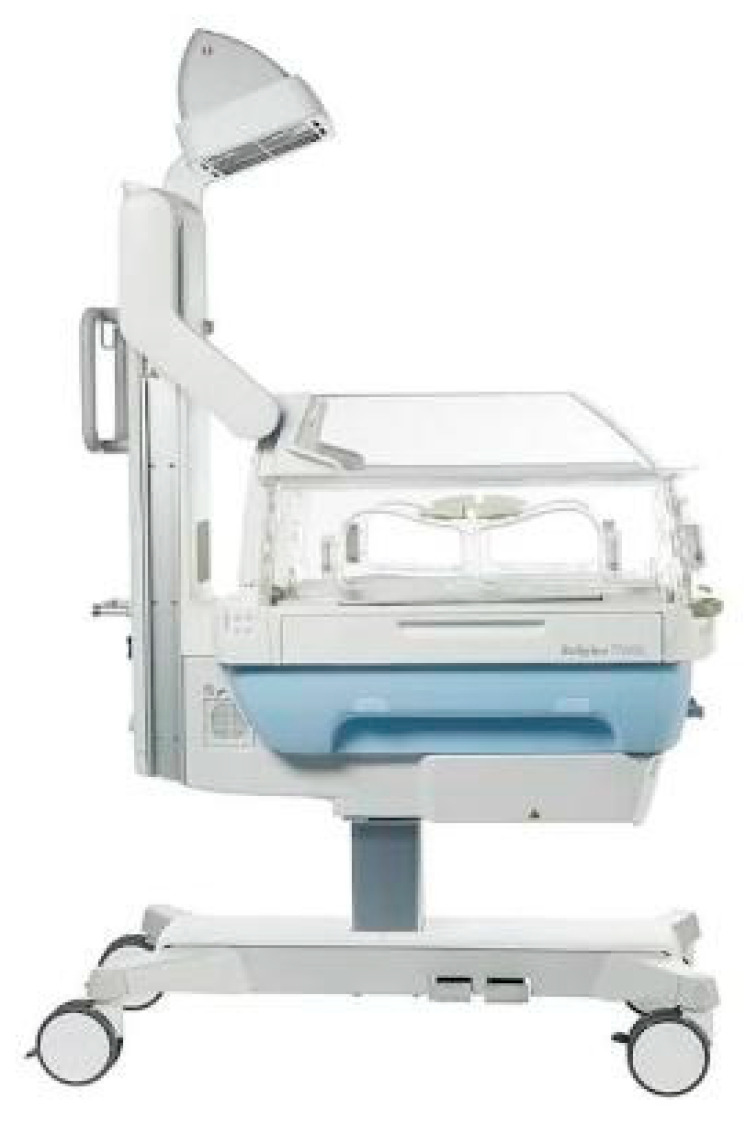
The Babyleo^®^ TN500. https://www.draeger.com/it_it/Hospital/Products/Thermoregulation-and-Jaundice-Management/Neonatal-Closed-Care/Draeger-Babyleo-TN500 (accessed on 6 February 2021).

**Figure 3 children-08-00334-f003:**
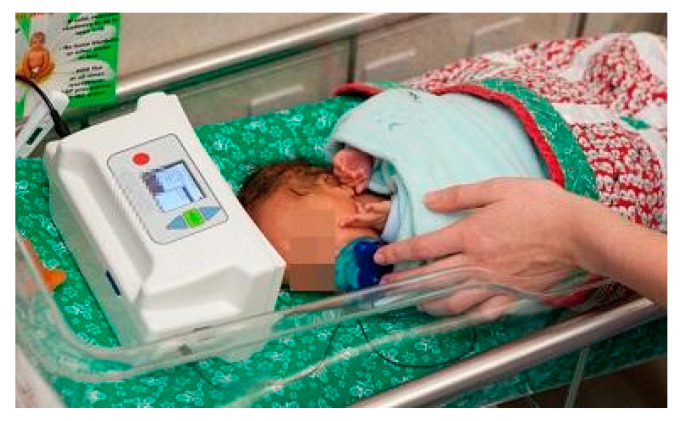
The Pacifier Activated Lullaby System. https://www.nurseshow.com/?filename=pacifier-activated-lullaby-system-a-real-PAL®-for-premature-babies (accessed on 6 February 2021).

**Figure 4 children-08-00334-f004:**
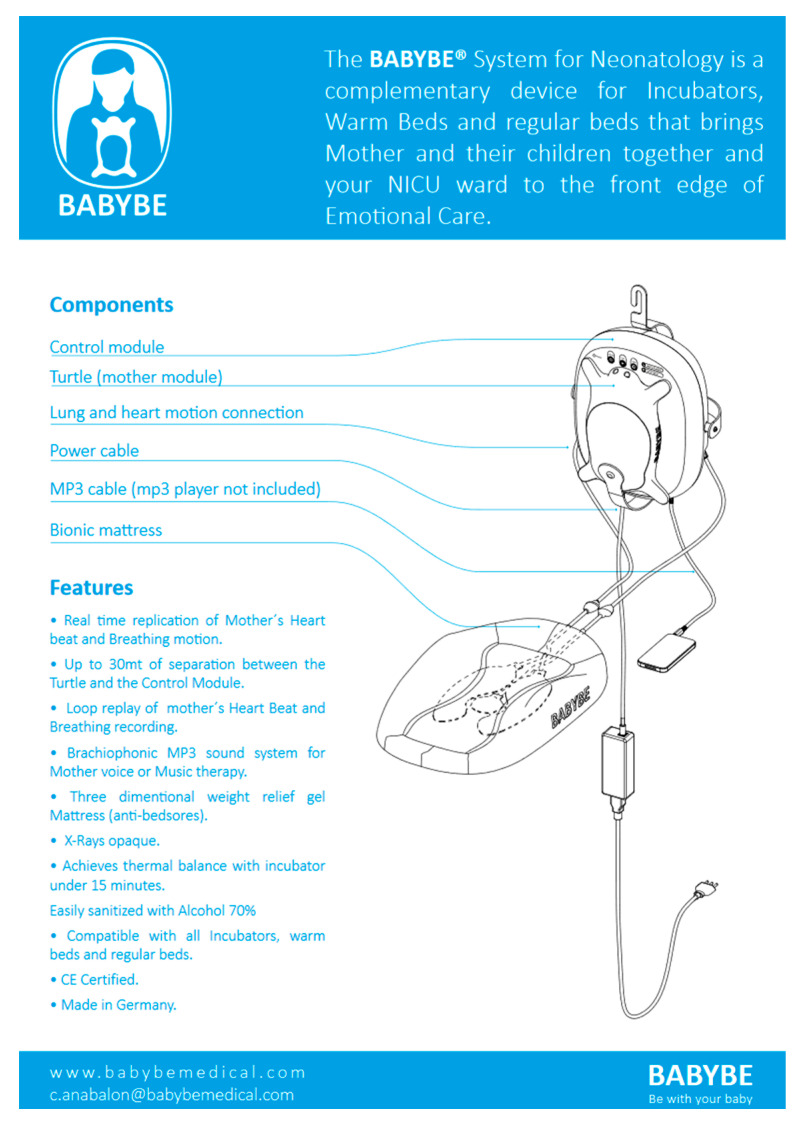
Babybe^®^ system. https://www.connect-medizintechnik.at/downloads/babybe-system-complete.pdf (accessed on 8 February 2021).

**Figure 5 children-08-00334-f005:**
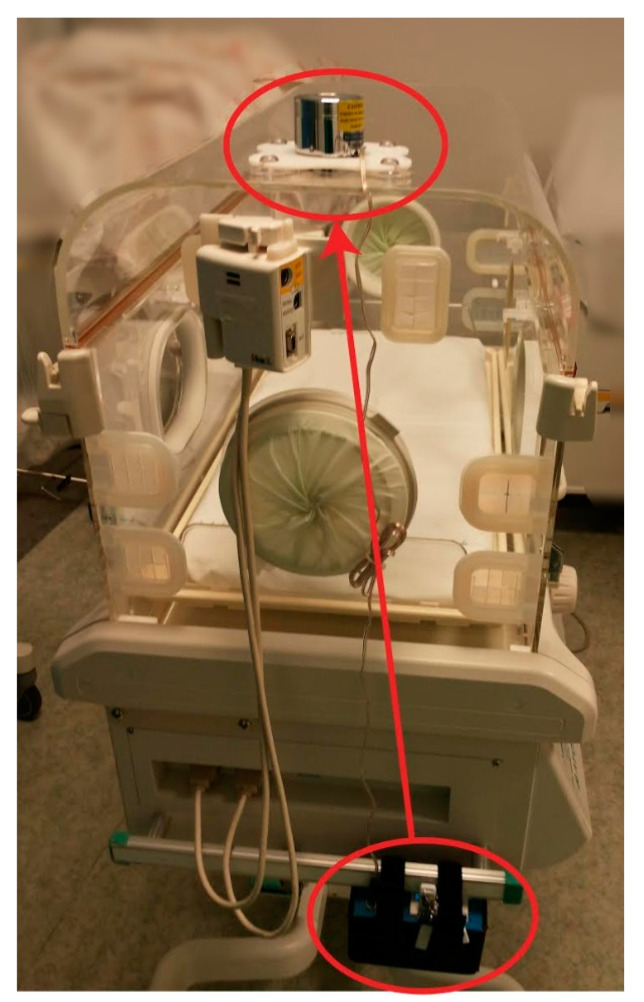
MAMI VOiCE system. In this figure we can see the amplifier attached to the incubator with a simple strap closure; it is then connected through a cable to the vibro-transducer on top.

**Figure 6 children-08-00334-f006:**
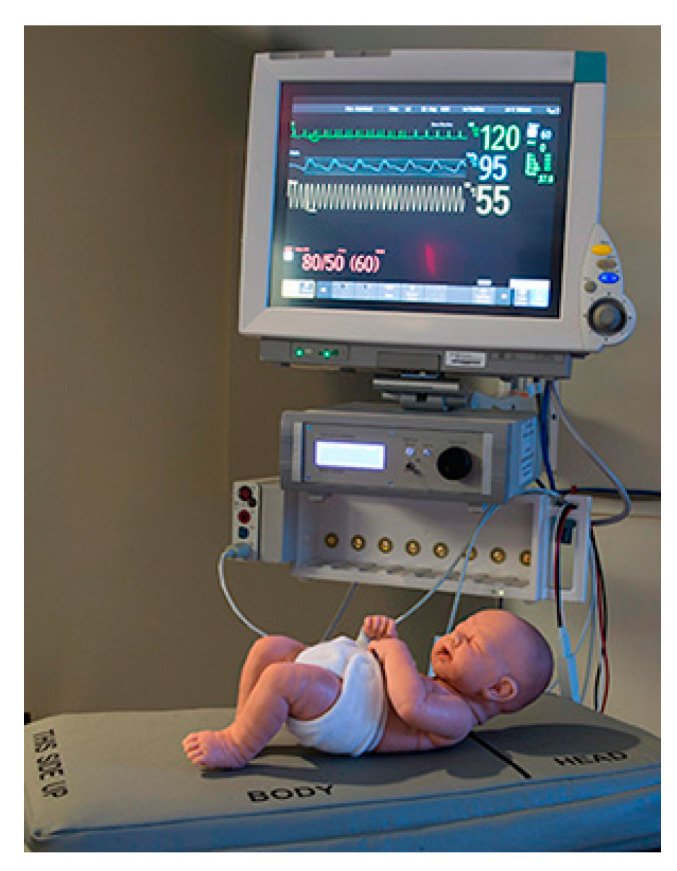
A 3D render by the Wyss Institute at Harvard University. https://wyss.harvard.edu/news/effectiveness-of-infant-apnea-prevention-technology-demonstrated-in-clinical-trial/ (accessed on 8 February 2021).

**Figure 7 children-08-00334-f007:**
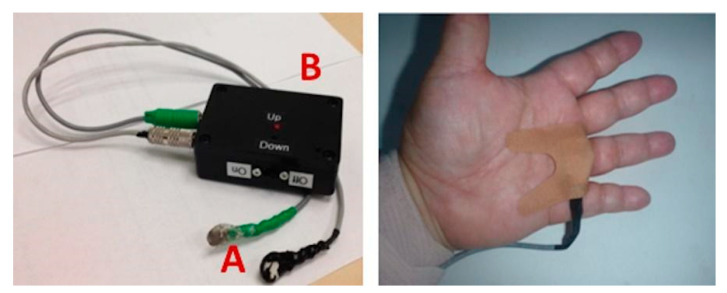
The stimulation device and the “puck”. A: small vibrating disks which are taped to the skin over proprioceptive fibers as in the figure on the right on the hand (or foot), B: a stimulation device, containing a low voltage battery that powers the vibration motor through flexible cables. https://journals.plos.org/plosone/article?id=10.1371/journal.pone.0157349 (accessed on 6 February 2021).

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
