# Peer review of "Sensory Stimulation in the NICU Environment: Devices, Systems, and Procedures to Protect and Stimulate Premature Babies"

_children, 2021, doi:10.3390/children8050334_

Round 1
Reviewer 1 Report
This paper called «narrative review» is an essay about sensory stimulation in the NICU environment. It covers many aspects and is based on a large body of references. It is well written but rather long. If space is limited, shortening should be possible without loss of content (suggestions below).
I realized that in this field there are very few randomized controlled trials to demonstrate the effect of a specific intervention. Therefore the authors are not able to provide statistically based data but rather ideas. There is one exception: for music therapy there are a number of controlled trials summarized in a review (Ref. 13). However, as these authors point out, they were not able to carry out a meta-analysis because type and timing of the interventions, the patient characteristics and the outcome measured were too heterogeneous.
The paper describes several interventions MAMI VOiCE, Babyleo, PAL (Pacifier Activated Lullaby System), Neoasis and Babybe® with potential physiological, neurodevelopmental or behavioral effects. However the evidence for these effects is far from being established. For most interventions there are only small pilot studies available.
In their essay, the authors select positive findings giving a biased view. For example the randomized trial about a pacifier-activated music player published in Pediatrics (Ref. 66) demonstrated a significantly better oral feeding rate and volume intake in the intervention group compared with the control group, but no difference in weight gain, cortisol levels and hospital stay (p=0.7).
The tenor of this paper is very optimistic. Given the lack of or weak proof of the effect of most devices, I miss a critical appraisal.
- P2, L57-72: This enumeration is a summary that could be dropped if space is scarce or put at the end.
- The Babybe System is described in extensor. It could be shortened by referring to previous publications (Ref 77 to 82). The same applies for the MAMI VOiCE (Ref. 83), the concept of neuromodulation (Ref 94-95), the Neoasis noise control device (Ref 51) and Babyleo (Ref 53)
- The number of references could be reduced. Examples: Ref 88 is cited in Ref 87. Ref 98 cited in Ref 95.
Specific comments:
- P4, L150: The text to figure 1 is not appropriate. The MRI scans are from a term infant not from a premature infant. 1 can be omitted.
- P 15, L 576: “all studies found no negative outcomes” most studies were not powered to exclude a negative effect.
- P15, L 594: Further limitations are: small sample sizes, different age (very preterm to tem infants)
- P15, L 598: not only these two effects need additional research, but almost all devices mentioned.
The conclusions are reasonable. I strongly agree with the statement that sophisticated devices do not replace the maternal presence.
Author Response
Thank you very much for having deeply revised the Manuscript ID Children-1132766, titled “Sensory stimulation in the NICU environment: devices, systems and procedures to protect and stimulate premature babies.”
We are also grateful to the reviewers for their thoughtful considerations.
We aimed to respond carefully to each reviewers comments and to the editorial recommendations.
Editorial Office's comments to authors:
It has been reviewed by experts in the field and we request that you make major revisions before it is processed further.
We have done the suggested major revision. We have clearly highlighted all the revisions, using the "Track Changes" function in Microsoft Word.
Reviewer #1: general comments:
This paper called «narrative review» is an essay about sensory stimulation in the NICU environment. It covers many aspects and is based on a large body of references. It is well written but rather long. If space is limited, shortening should be possible without loss of content (suggestions below).
I realized that in this field there are very few randomized controlled trials to demonstrate the effect of a specific intervention. Therefore the authors are not able to provide statistically based data but rather ideas. There is one exception: for music therapy there are a number of controlled trials summarized in a review (Ref. 13). However, as these authors point out, they were not able to carry out a meta-analysis because type and timing of the interventions, the patient characteristics and the outcome measured were too heterogeneous.
The paper describes several interventions MAMI VOiCE, Babyleo, PAL (Pacifier Activated Lullaby System), Neoasis and Babybe® with potential physiological, neurodevelopmental or behavioral effects. However the evidence for these effects is far from being established. For most interventions there are only small pilot studies available.
We have researched the term “narrative review” and come to the conclusion that it could still be appropriate, due to the nature of our work being nonsystematic and non-empirical, but rather indeed a “scientific narration”
In their essay, the authors select positive findings giving a biased view. For example the randomized trial about a pacifier-activated music player published in Pediatrics (Ref. 66) demonstrated a significantly better oral feeding rate and volume intake in the intervention group compared with the control group, but no difference in weight gain, cortisol levels and hospital stay (p=0.7).
The tenor of this paper is very optimistic. Given the lack of or weak proof of the effect of most devices, I miss a critical appraisal.
The Introduction, Discussion and Conclusion have been critically modified. All the devices have been condensed where possible.
P2, L57-72: This enumeration is a summary that could be dropped if space is scarce or put at the end.
This part has been removed
The Babybe System is described in extensor. It could be shortened by referring to previous publications (Ref 77 to 82). The same applies for the MAMI VOiCE (Ref. 83), the concept of neuromodulation (Ref 94-95), the Neoasis noise control device (Ref 51) and Babyleo (Ref 53)
We have shortened the descriptions of the above devices
The number of references could be reduced. Examples: Ref 88 is cited in Ref 87. Ref 98 cited in Ref 95.
The suggested references have been deleted
Specific comments:
P4, L150: The text to figure 1 is not appropriate. The MRI scans are from a term infant not from a premature infant. 1 can be omitted.
Figure 1 has been omitted.
P 15, L 576: “all studies found no negative outcomes” most studies were not powered to exclude a negative effect.
P15, L 594: Further limitations are: small sample sizes, different age (very preterm to tem infants)
P15, L 598: not only these two effects need additional research, but almost all devices mentioned.
The conclusions are reasonable. I strongly agree with the statement that sophisticated devices do not replace the maternal presence.
Reviewer 2 Report
Thank you for the opportunity to review the manuscript titled Sensory stimulation in the NICU environment: devices, systems and procedures to protect and stimulate premature babies. This is interesting work that provides a synthesis of possible interventions to enhance outcomes for premature babies admitted to the NICU.
There is some inconsistency with how the work is referred to. Initially it is referred to as a study, but later highlighted as a narrative review. It needs to be made clear that this is indeed a review article and this language needs to be consistent throughout the paper. The manuscript requires some clarity around the methods used to select literature and explore the included technological devices, systems and techniques. There is also some confusion around whether this is a review of previous studies, or of products. If this is a review of the literature, a search strategy including databases searched, key search terms and inclusion and exclusion criteria needs to be included. Also a PRISMA diagram and summary table would be required. The methods and results require separate sub-headings and the results need to commence with a summary of what exactly has been reviewed.
The authors need to consider whether section 3 NICU environment is actually part of the results or whether it would be better placed in the introduction to provide context.
The results are lengthy, mainly descriptive and could be substantially condensed. The use of a table that summarised the different technological devices, systems and techniques may be helpful and enhance readability of the manuscript.
p. 2 line 82 refers to numerous 'researches' which should read numerous studies.
The discussion requires further development. Currently it consists of a series of 1 or 2 sentence paragraphs rather than a well developed critical discussion.
Author Response
Thank you very much for having deeply revised the Manuscript ID Children-1132766, titled “Sensory stimulation in the NICU environment: devices, systems and procedures to protect and stimulate premature babies.”
We are also grateful to the reviewers for their thoughtful considerations.
We aimed to respond carefully to each reviewers comments and to the editorial recommendations.
Editorial Office's comments to authors:
It has been reviewed by experts in the field and we request that you make major revisions before it is processed further.
We have done the suggested major revision. We have clearly highlighted all the revisions, using the "Track Changes" function in Microsoft Word.
Reviewer #2: general comments:
Thank you for the opportunity to review the manuscript titled Sensory stimulation in the NICU environment: devices, systems and procedures to protect and stimulate premature babies. This is interesting work that provides a synthesis of possible interventions to enhance outcomes for premature babies admitted to the NICU.
There is some inconsistency with how the work is referred to. Initially it is referred to as a study, but later highlighted as a narrative review. It needs to be made clear that this is indeed a review article and this language needs to be consistent throughout the paper. The manuscript requires some clarity around the methods used to select literature and explore the included technological devices, systems and techniques. There is also some confusion around whether this is a review of previous studies, or of products. If this is a review of the literature, a search strategy including databases searched, key search terms and inclusion and exclusion criteria needs to be included. Also a PRISMA diagram and summary table would be required. The methods and results require separate sub-headings and the results need to commence with a summary of what exactly has been reviewed.
We are currently working to highlight the nature of our work; with narrative review we intend a different approach than the systematic one (that as you’ve mentioned include PRISMA guidelines and a database of the search engine used). We specifically used the term narrative because it provided us a way of presenting scientific findings that currently don’t have that body of literature and empirical evidence that could allow us to conform to PRISMA guidelines and to produce comparative results. Given the narrowness of the field and of the patients most of the studies are in the “alpha” stage or are referred to population that can’t be used as a “reference sample” also because of the heterogeneity / differences between: infants health, device used, medical procedures applied, hospitals guidelines, parents’ mental health and so on. Thus, for each device we’ve presented we have analyzed specific studies with specific population and methods, each device is representative of a/more studies as required by the Special Issue guidelines and editors: section (e) bio-engineering and technological innovation to enhance the NICU soundscape. https://www.mdpi.com/journal/children/special_issues/Sound_Neonatal_Intensive_Care_Unit_NICU
We were specifically asked of reviewing the studies about technological devices and techniques to conform to the Special Issue topic as stated in the Introduction:
The purpose of this narrative review was to present and investigate the available technological devices, systems and techniques used to promote the premature babies’ correct development, and protect them from the excessive exposure to sounds and lights. The effect of such devices on parents and medical staff was also investigated, where possible. The study covered different areas of application of the vibration/sound treatments, many of them administered during music therapy sessions
Specific comments:
The authors need to consider whether section 3 NICU environment is actually part of the results or whether it would be better placed in the introduction to provide context.
Section 3 has been eliminated.
The results are lengthy, mainly descriptive and could be substantially condensed. The use of a table that summarised the different technological devices, systems and techniques may be helpful and enhance readability of the manuscript.
A table of content has been made
- 2 line 82 refers to numerous 'researches' which should read numerous studies.
The term “researches” has been changed with “studies”
The discussion requires further development. Currently it consists of a series of 1 or 2 sentence paragraphs rather than a well developed critical discussion.
The results have been condensed where possible with major revisions in the introduction, discussion and conclusion.
Round 2
Reviewer 2 Report
Thank you for the opportunity to review this revised manuscript. The authors have addressed all reviewers' comments and the manuscript has been improved substantially.
Author Response
We sincerely thank you for your attention and for the previuos
revisions you proposed to improve our work.
Francesco, Carmen, Prof. Chirico